

# Mechanism of *Brevibacillus brevis* strain TR-4 against leaf disease of *Photinia × fraseri* Dress

Chenxinyu Ji[1], Yun-Fei Li[2], Yao Yao[1], Zengrui Zhu[1] and Shengfeng Mao[1,3]

[1] Nanjing Forestry University, Nanjing, Jiangsu, China
[2] Jinling Institute of Technology, Nanjing, Jiangsu, China
[3] Zhejiang Agriculture and Forestry University, Hangzhou, Zhejiang, China

## ABSTRACT

**Background**. *Colletotrichum* species are among the most common pathogens in agriculture and forestry, and their control is urgently needed.

**Methods**. In this study, a total of 68 strains of biocontrol bacteria were isolated and identified from *Photinia × fraseri* rhizosphere soil.

**Results**. The isolates were identified as *Brevibacillus brevis* by 16S rRNA. The inhibitory effect of TR-4 on *Colletotrichum* was confirmed by an *in vitro* antagonistic experiment. The inhibitory effect of TR-4 was 98% at a concentration of 10 µl/ml bacterial solution, protection of the plant and inhibition of *C. siamense* was evident. Moreover, the secretion of cellulase and chitosan enzymes in the TR-4 fermentation liquid cultured for three days was 9.07 mol/L and 2.15 µl/mol, respectively. Scanning electron microscopy and transmission electron microscopy confirmed that TR-4 destroyed the cell wall of *C. siamense*, resulting in leakage of the cell contents, thus weakening the pathogenicity of the bacteria.

## INTRODUCTION

*Photinia × fraseri* Dress, a small evergreen tree or shrub, prefers warm, moist environments and exhibits vibrant colors in direct light (*Toscano et al., 2016*). Predominantly found in Southeast Asia, Eastern Asia, and North America, it is widely cultivated across various provinces in China for garden greening. The plant's disease susceptibility has important economic implications (*Li et al., 2023*). In 2019, a major disease of *P. × fraseri* was found in Nanjing, Jiangsu Province, China (*Mao et al., 2020*). In the early stage of infection, the infected leaves exhibited small, round, light reddish-brown spots that gradually expanded to round areas, with light gray centers and brown edges (*Chen et al., 2021*). After a series of verification steps, the disease affecting the plants was identified as being caused by the fungal genus *Colletotrichum*. This genus is known for attacking the roots, stems, leaves, flowers, and fruits of various plants globally, leading to decreased agricultural product quality and substantial economic losses (*Cao et al., 2021*). In recent years, *Colletotrichum* has been found on a variety of crops and plants in many places around the world. For example, *Colletotrichum* was found on holly in Zhejiang Province, China, in 2018 (*Feng et*

Corresponding author
Shengfeng Mao, maosf@126.com

*al., 2023*), and on Litchi in Guangzhou Province in 2020 (*Ling et al., 2019*). Research on the prevention and control of *Colletotrichum* infection is urgently needed.

In recent years, biological control has attracted widespread attention due to its environmental friendliness and safety (*Aggeli et al., 2020*). Scientists have focused on the use of antagonists and their active substances. In 1996, *Brevibacillus*, a rod-shaped gram-positive bacterium, was established as a separate genus (*Goto et al., 2004*). In previous studies, *Brevibacillus* was shown to be a widely effective biocontrol bacterium. It also has an inhibitory effect on many resistant fungi (*Arumugam et al., 2018*) and can also control some hymenopterans (*Babar et al., 2022*). *B. brevis*, as a biocontrol strain, has great research potential in different fields.

*Brevibacillus* species are omnipresent in agricultural soils and can secrete structurally diverse secondary metabolites with broad antibiotic spectra (*Yang & Yousef, 2018*). *Brevibacillus* spp. are among the PGPR (plant growth promoting rhizobacteria) groups used as biofertilizers or biopesticides on different crops and against a variety of soil-borne and foliar pathogens (*Devi et al., 2019*). Using genome mining, many antimicrobial compounds, such as those produced by *Brevibacillus* and antimicrobial cyclic lipopeptides, which are found in *Brevibacillus laterosporus,* were discovered (*Rasool Kamli et al., 2022*). Complete genome sequencing technology has good application prospects for the discovery of genome sequence information for unknown bacteria and the exploration of critical functional genes (*Schuch et al., 2016*).

In this study, we sampled the rhizosphere soil of healthy *P. × fraseri* plants, screened the soil bacteria, and obtained a bacterium with excellent biocontrol efficacy. After the study in this paper, it was determined that TR-4 had an excellent inhibitory effect on *Colletotrichum*, and the TR-4 was identified as *Bacillus brevis*. This study further investigated the control of *C. siamense* by *B. brevis* and laid the groundwork for future research on whether *B. brevis* can colonize *P. × fraseri*. The biocontrol ability of *B. brevis* was determined from the aspect of endogenous hormones, and the results showed that TR-4 was a biocontrol bacterium with excellent inhibition against Colletotrichum pathogens.

## MATERIAL AND METHODS

### Experimental material

The *C. siamense* strain was obtained from the Laboratory of Forest Protection, Nanjing Forestry University, Nanjing, Jiangsu Province, China, and was stored in the China Forestry Microbial Strain Preservation and Management Center under the preservation number CFCC54215.

The strain was cultured on PDA medium and subsequently cultured in an incubator at 25 °C. The bacterial strains were isolated from healthy *Photinia rubra* rhizosphere soil, cultured on NA medium and subsequently cultured in a 30 °C incubator.

The TR-4 fermentation liquid broth was the bacterial broth of TR-4 added to 100 ml of LB liquid medium and incubated at 30 °C for 3 days with an $OD_{600}$ of about 5.

The *P. × fraseri* leaves used in the experiment were obtained from Yaping Nursery in Nanjing, two-year-old seedlings, which were transplanted and cultured at 28 °C under natural light.

## Isolation and screening of antagonistic bacteria

A total of 50 g of soil was taken from five randomly selected points in the inter-root soil of healthy *P. × fraseri* plants by random sampling method and diluted using sterile water to obtain dilutions at concentrations of $10^{-1}$ $10^{-2}$, $10^{-3}$, $10^{-4}$ and $10^{-5}$, respectively (*Ambardar & Vakhlu, 2013*). The dilutions were spread on LB solid medium by spreading method and incubated in an incubator at 30 °C for 3 days. Single strains were isolated after labeling based on colonies with different morphological and color characteristics. Counts were taken and the inhibitory effect of the isolated antagonistic bacteria on *C. siamense* was determined using the plate antagonism method. The antagonistic effect on fungal mycelium was calculated as percentage growth inhibition (% GI). The formula for growth inhibition was 1-(experimental/control) × 100%. Data were obtained from three different experiments.

## Effect of TR-4 fermentation liquid broth on the germination of *C. siamense* spores

A total of 500 µl of 0.1% glucose aqueous solution was added to the *C. siamense* spores suspension (spore suspension concentration is $10^6$/ml) and TR-4 fermentation liquid in a 2 ml aseptic centrifuge tube, and the concentration was adjusted to the $EC_{50}$ (median effective concentration), 10 $EC_{50}$, and $EC_{90}$ according to the ratio, with a final volume of 500 µl. LB liquid medium was used instead of TR-4 bacterial solution as a control. The spores were cultured in a dark incubator at 25 °C, after which 5 µl was extracted from the test tube every 12 h and placed on a glass plate, until the control spores had fully germinated. Spore germination was observed under a Zeiss microscope.

## *In vivo* antagonism experiment

The experimental samples was divided into three groups. The first group was the biocontrol group. The spore solution of *C. siamense* 10 µl (spore suspension concentration of $10^6$/ml) was inoculated first, and the bacterial solution of TR-4 fermentation liquid was inoculated 24 h later to observe the control effect of TR-4 on *C. siamense* on plant leaves of *P. × fraseri*. In second group (the protection group), plants were sprayed with TR-4 fermentation liquid; they were completely dried and inoculated with *C. siamense* spore solution to observe whether TR-4 could help plants resist *C. siamense* infection on the leaves. The third group (the control group) was inoculated with only the *C. siamense* spore solution, and each experiment was repeated 3 times.

## Molecular identification of TR-4

The universal primers 27F (5′-AGAGTTTGATCCTGGCTCAG-3′) and 1492R (5′-CGGCTACCTTGTTACCAC-3′) of the bacterial 16S rRNA gene were used for PCR amplification (*Johnson, Bowman & Dunlap, 2020*). The PCR mixture was as follows: 2 × Taq PCR Master Mix 25 µL, 2 µL of F primer, 2 µL of R primer (the primer working solution concentration was 10 µM), 2 µL of template DNA (DNA extraction was performed using Vazyme's DNA extraction kit), and ddH$_2$O to a total volume of 50 µL. The PCR procedure was as follows: predenaturation at 94 °C for 5 min; denaturation at 94 °C for 1 min; annealing at 58 °C for 1 min; and denaturation at 72 °C for 2 min. Thirty cycles

were repeated and finally extended for 10 min at 72 °C. The PCR products were sequenced by Nanjing Bioengineering. After NCBI BLAST comparison, MEGA7 software was used for sequence analysis, and the neighbor-joining (NJ) method was used to construct a phylogenetic tree.

### Analysis of TR-4 secretions
### Determination of ferriphilin

Iron is a micronutrient widely found in the Earth's crust; a small amount of iron is necessary for plants, and iron deficiency is a plant nutrient disorder. Iron forms iron oxide hydrates in the environment, resulting in a lower concentration of free iron and reduced bioavailability. The CAS test solution is a bright blue compound consisting of chromium, cetyltrimethyl ammonium bromide, and iron ions. When the iron ions in the blue test solution are removed by the ferritin secreted by microorganisms, the CAS test solution changes from blue to orange, so the CAS liquid medium can be used to detect the production of ferritin by microorganisms (*Pérez-Miranda et al., 2007*). The light absorption (As) of the supernatant after centrifugation was measured at 630 nm and adjusted to zero using double steaming water as a control. Another blank medium was mixed with the CAS test solution in equal amounts, and its light absorption value was taken as the reference ratio (Ar). The experimental method was performed according to the CAS assay kit instructions.

### Determination of cellulase activity (3, 5-dinitrosalicylic acid method)

Cellulase hydrolyzes cellulose to produce cellobiose, glucose and other reducing sugars, which can reduce the nitro in 3, 5-dinitrosalicylic acid to orange amino compounds, and use a colorimetric method to determine the generation of reducing compounds to indicate the activity of the enzyme (*Song et al., 2016*). The experimental method refers to the assay of cellulase by Song et al.

### Determination of chitosanase activity

The modified Schales method was used to determine the enzyme activity. The principle behind this process is that soluble chitosan undergoes enzymolysis and releases reducing sugars, which react with the Schales reagent to change color. With N-acetylglucosamine as the standard sugar, the light absorption value of the reducing sugars was determined *via* a spectrophotometer at 420 nm. The amount of enzyme that breaks down 1 μ/mol NAG per minute is defined as one unit of activity (U) (*Mojumdar et al., 2019*).

### Microscopic analysis of the inhibitory effect of strain TR-4 on *C. siamense*

Scanning electron microscopy (SEM) and transmission electron microscopy (TEM) samples were obtained from fresh PDA plates. The samples were divided into two categories:(1) no inhibition on fungal growth, such as control group; (2) obvious inhibition of fungal growth, such as the experimental group.

The samples were fixed with 2.5% glutaraldehyde and 1% cesium tetroxide at room temperature, dehydrated with ethanol, critically dried, covered with gold, and observed with a scanning electron microscope (JEM 2100) .

The TEM samples were prepared by a similar method. The *C. siamense* are cut into 2 × 3 mm slices. The specimens were placed in 2.5% glutaraldehyde solution, fixed at room temperature for 5∼6 h, cleaned with 0.1M phosphate buffer (PBS, pH 7.2) for 5 times, fixed with 1% cesium tetroxide for 1.5 h, cleaned with PBS for eight times, dehydrated with ethanol and acetone, coated with SPUS resin, sliced 10 μm, stained, and observed under transmission electron microscope.

## Data analysis

In this research, all the experiments were carried out in triplicate and repeated three times to get accurate and reliable data. A completely randomized design was used for the greenhouse experiment, and its data were examined with analysis of variance (ANOVA) followed by the least significant difference (LSD) tests at $p < 0.05$ using the DPS v9.5 statistical software package.

# RESULTS

## Soil sample collection and screening of antagonistic bacteria

The experiment isolated 68 soil bacteria from five different concentrations of soil solutions. After the antagonism experiment of the 68 isolated bacteria against *C. siamense*, it was concluded that a total of 13 strains of bacteria had an effect on *C. siamense*. The strain with the strongest inhibitory effect was selected as the experimental strain and named TR-4(Fig. 1).

## *In vitro* antagonistic experiment and inhibitory effects of TR-4 on *C. siamense* spore germination

To further determine the inhibitory effect of TR-4 on *C. siamense*, the concentrations used were set to 0.001 μl/ml, 0.01 μl/ml, 0.1 μl/ml, 1 μl/ml and 10 μl/ml through *C. siamense* culture and experiments (Fig. 2). According to DPS v9.5 analysis, the average $EC_{50}$ was 0.0022 ±0.0013 μl/ml, and the average $EC_{90}$ was 0.8115 ±0.1024 μl/ml (Table 1).

Moreover, to verify whether the TR-4 strain has an antagonistic effect on *C. siamense* spore germination, *C. siamense* spores were treated with sterile TR-4 filtrate, and the final solution $EC_{50}$, 10 $EC_{50}$ and $EC_{90}$ were determined. We found that the spore germination rate of *Bacillus anthracis* treated with the TR-4 strain fermentation filtrate was significantly lower than that of the control group within 12 h. After 48 h, the spore germination rate of the control was 98%, while the spore germination rate of the TR-4 strain treated with fermentation filtrate was significantly lower than that of the control (Fig. 3 and Table 2). The mycelial length and mycelial branching number after spore germination were significantly lower than those in the control group, and the differences did not decrease with increasing time. Like in other studies, in this study, TR-4 was shown to reduce the germination rate of *C. siamense* and thus reduce their pathogenicity.

## *In vivo* antagonism experiment

The results of *in vivo* antagonistic experiments revealed that TR-4 has significant inhibitory effects on *C. siamense* and protective effects on plants. In the first group of control experiments, the lesion did not expand after spraying the TR-4 bacterial solution, and

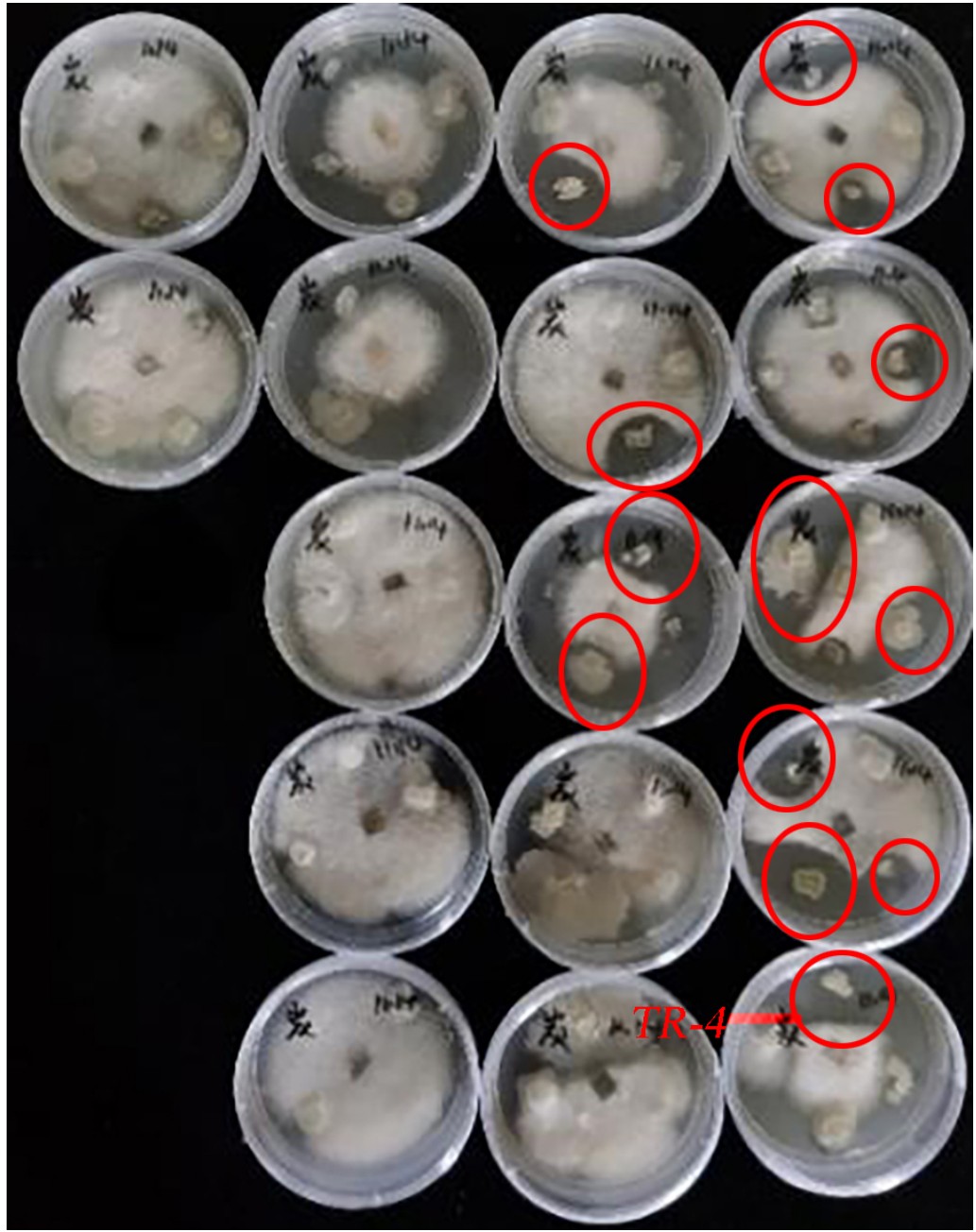

**Figure 1** **Plate antagonism experiments between soil bacteria and *C. siamense*.** Thirteen strains of bacteria inhibited *C. siamense*, with the largest circle of inhibition being TR-4.

the control effect was remarkable. In the second group of plant protection experiments, compared with those in the first group of experiments, although the effects were not ideal, some inhibitory effects were detected. The results showed that TR-4 can also be directly inhibit *C. siamense* growth on leaves of *C. siamense* plants (Fig. 4). In this study, the phenotype and severity of the disease in the biocontrol group were significantly lower than

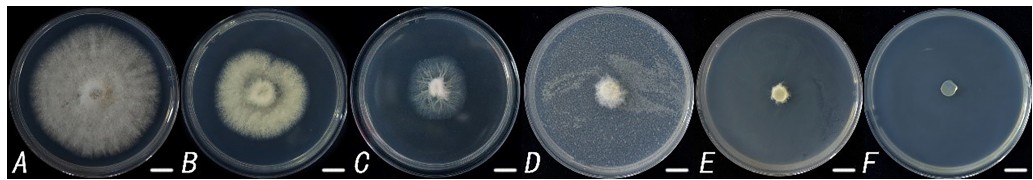

**Figure 2** **Inhibitory effect of different concentrations of TR-4 fermentation broth on *Colletotrichum siamense*.** The concentrations of TR-4 fermentation broth in B-F were 0.001 μl/ml, 0.01 μl/ml, 0.1 μl/ml, 1 μl/ml and 10 μl/ml, respectively, and A was the control; Scale bars = 1 cm.

**Table 1** **Colony size and inhibition rate of *Colletotrichum siamense* after treatment with different concentrations of TR-4 fermentation broths.** Bacteriostatic rate of fermentation broth to *Colletotrichum siamense* at different concentrations of TR-4.

| Treatment concentration (μl/mL) | Colony diameter (mm) | Inhibition rate (%) |
|---|---|---|
| Control | 38.2867 ±0.1242[a] | 0 |
| 0.001 | 23.4933 ±0.2122[b] | 44.88 |
| 0.01 | 21.1433 ±0.1415[c] | 52.82 |
| 0.1 | 10.2967 ±0.1079[d] | 86.85 |
| 1 | 8.6000 ±0.1539[e] | 91.25 |
| 10 | 7.3700 ±0.0557[f] | 95.75 |

**Notes.**

The difference of lowercase letters a and b in the table indicates that the bacteriostatic rate is significantly different at the level of $P < 0.05$.

those in the control group, indicating that TR-4 inhibited the incidence of *C. siamense* in *P. × fraseri*.

## Molecular identification of TR-4

The 16S rRNA sequences of TR-4 were analyzed. PCR amplification and sequencing revealed that the length of the 16S rRNA gene was 1430 bp. The 16S rRNA nucleotide sequence of TR-4 was registered in the GenBank database under accession number OP658963.1. The 16S rRNA sequence of TR-4 was identified by the National Center for Biotechnology Information (NCBI) database and shared 99% homology with the 16S rRNA gene sequence of *B. brevis* ON014586. A phylogenetic tree was constructed using MEGA 6 software, and strain TR-4 was identified as *B. brevis* (Fig. 5).

## Analysis of TR-4 secretions

In the determination of ferritin, according to the radical formula [(AR-AS)/Ar] × 100%, the relative content of ferritin was 88.46 ±2.08%.

In the determination of cellulase activity, the absorbance of the color developing solution was determined, and a standard curve was drawn with the absorbance as the vertical coordinate and the glucose content as the horizontal coordinate. The linear regression equation was used to construct the standard curve of the reducing sugars. The standard curve equation was $y = 0.16822x + 0.0073$ ($R^2 = 0.8802$). The absorbances of the blank

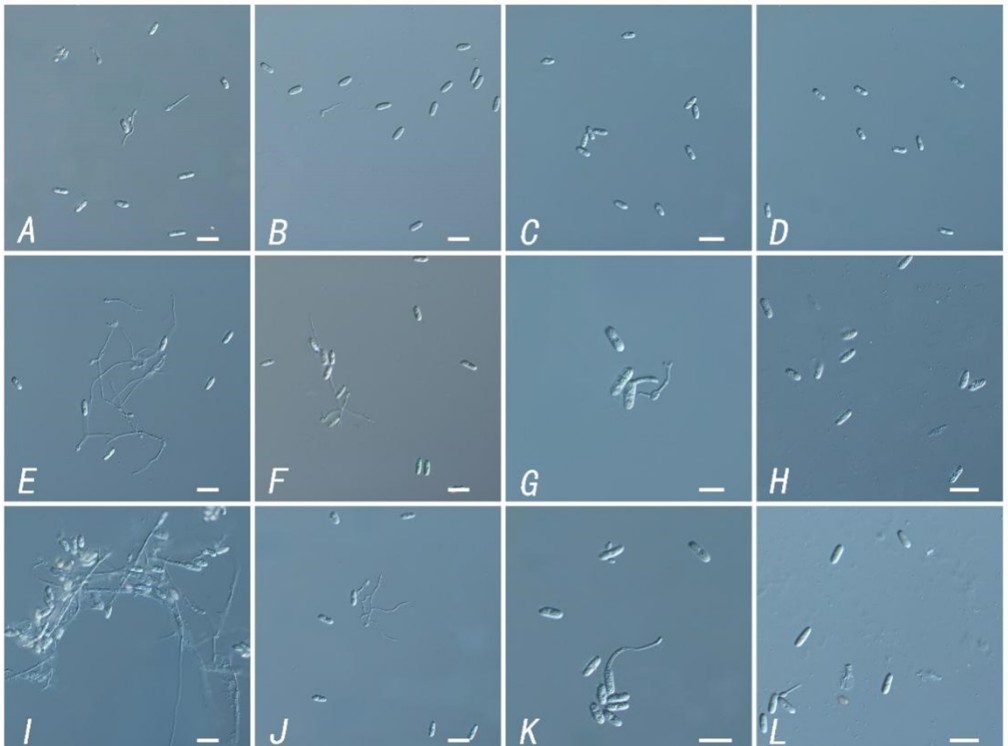

**Figure 3 Inhibitory effect of different concentrations of TR-4 fermentation broth on the germination of *Colletotrichum siamense* spores.** (A) Control of spore germination at 12 h; (B) spore germination with the concentration of $EC_{50}$ in fermentation broth at 12 h; (C) spore germination with a fermentation broth concentration of $10EC_{50}$ at 12 h; (D) spore germination with the concentration of $EC_{90}$ in the fermentation broth at 12 h; (E) control of spore germination at 24 h; (F) spore germination with the concentration of $EC_{50}$ in fermentation broth at 24 h; (G) spore germination with a fermentation broth concentration of $10EC_{50}$ at 24 h; (H) spore germination with the concentration of $EC_{90}$ in the fermentation broth at 24 h; (I) control of spore germination at 36 h; (J) spore germination with the concentration of $EC_{50}$ in fermentation broth at 36 h; (K) spore germination with a fermentation broth concentration of $10EC_{50}$ at 36 h; (L) spore germination with the concentration of $EC_{90}$ in the fermentation broth at 36 h; Scale bars = 10 μm.

**Table 2 Germination of *Colletotrichum siamense* spores after treatment with different concentrations of fermentation broth TR-4 at different time periods.** Effects of different concentrations of antagonistic antibiotic fermentation broth on spore germination.

| Germination time | Germination rate | | | |
|---|---|---|---|---|
| | Control | EC50 | 10EC50 | EC90 |
| 12 h | 30% | 6% | 0 | 0 |
| 24 h | 80% | 24% | 16% | 2% |
| 36 h | 98% | 32% | 24% | 8% |

tube and sample were 0.0264 and 0.0485, respectively. The above standard curve equations for the glucose content of the blank tube and sample were 0.1135 and 0.2449, respectively.

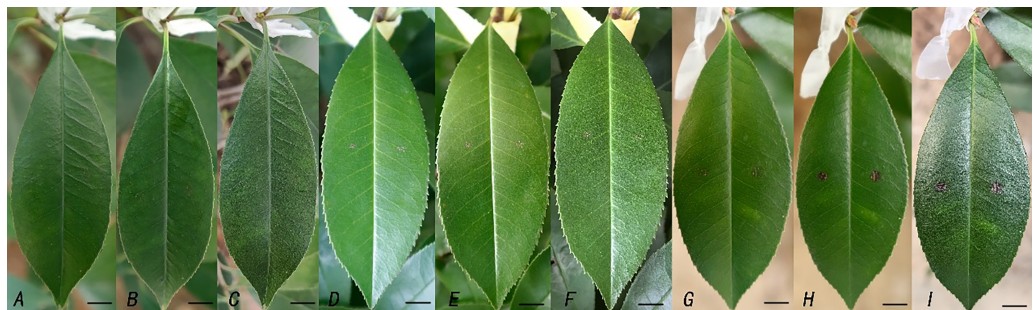

**Figure 4  Interaction between TR-4 and *Colletotrichum siamense in vivo.*** A, B and C were the leaf states of group 1, Group 2 and group 3 on day 3, respectively. D, E and F were the states of leaves in group 1, Group 2 and group 3 on day 5, respectively; G, H and I are the states of group 1, Group 2 and Group 3 on day 7, respectively.; Scale bars = 1 cm.

The results showed that strain TR-4 could produce cellulase, and the ability to produce cellulase was very strong.

In the chitosan enzyme activity assay experiment, according to the experimental method, the results were obtained as 2.15 μmol of reducing sugars and 0.215 U of enzyme activity units.

## Microscopic analysis of the inhibitory effect of strain TR-4 on *C. siamense*

Under scanning electron microscopy, normal mycelia of *C. siamense* were found to be evenly distributed, smooth and full. After treatment with TR-4, mycelial growth was abnormal, the mycelial shape and surface were deformed, and the surface was contracted. Under transmission electron microscopy, mitochondria, ribosomes, vacuoles, cell walls and even plasma from normal *C. siamense* cells were clearly visible. After treatment with the TR-4 strain, the cells of *C. siamense* exhibited obvious changes and damage. The cell wall was transparent, the organelles disappeared, and the vacuoles were deformed (Fig. 6).

## DISCUSSION

Prior to this, there have been many studies showing that *B. brevis* is a good biocontrol bacterial strain. In terms of the ability of rice biocontrol bacteria to control various rice diseases, among the 11 potential biocontrol bacteria, the best was *B. brevis* strain 1Pe2 (*Yang et al., 2007*). According to previous studies, *B. brevis* has a significant effect on tea tree *Gloeosporium-sinae-sinensis*, *Elsinoe leucospira*, *Phyllosticta theaefolia*, *Fusarium* sp. *Cercospora theae* and other pathogens, indicating that it has inhibitory effects on many *C. siamense* and is a biocontrol strain of great research value (*Yang et al., 2023*). It was proven that *B. brevis* has a broad spectrum.

In another experiment, it was suggested that yeast and non-viticultural yeasts inhibited fungal mycelia growth through metabolites, laminaria polysaccharide enzymes, nutrient competition, fungal spore germination inhibition, bud tube length shortening, and

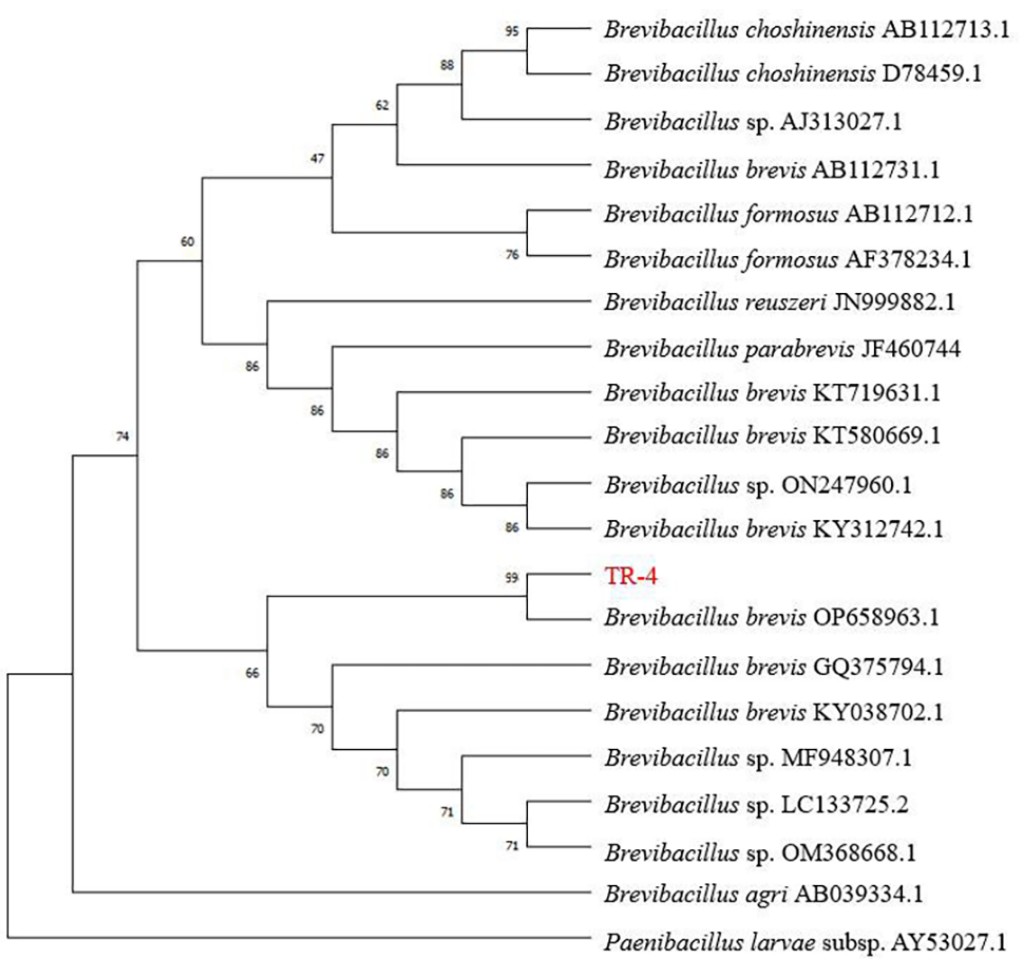

**Figure 5** **Neighbour-joining phylogenetic trees based on 16S rRNA gene sequences of *BreviBacillus brevis.*** Genetic distances were computed by Kimura's two-parameter model. Only bootstrap percentages above 50 % are shown.

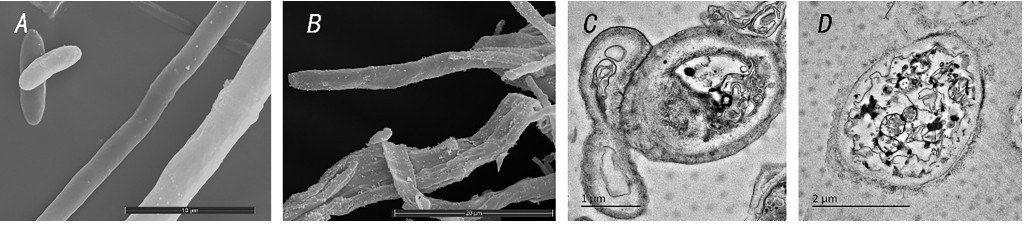

**Figure 6** **Microscopic observation of *Colletotrichum siamense* after control with TR-4.** A shows the state of *Colletotrichum siamense* under scanning electron microscopy; B is the state of *Colletotrichum siamense* under scanning electron microscope after TR-4 treatment; C is the state of *Colletotrichum siamense* under transmission electron microscopy; D is the state of *Colletotrichum siamense* after TR-4 treatment under projection electron microscopy. Scale bar: A = 10 μm, B = 20 μm, C = 1 μm, D = 2 μm.

antifungal volatiles (*Hilal et al., 2016*; *Nally et al., 2015*). Therefore, the spore germination test was used to verify the efficacy of the biocontrol bacteria.

The biocontrol activity of TR-4 on *C. gloeosporioides* on ripe olive fruits was verified. Biocontrol bacteria reduce the incidence and severity of *C. gloeosporioides*, and its incidence on fruit can be reduced by 50–90% (*Pesce et al., 2018*).

Iron is an essential element for the growth of *C. siamense*. Siderophiles produced by *B. brevis* can prevent the absorption of iron by *C. siamense*. As an index of biocontrol bacteria (*Wang et al., 2020*). In the natural environment, $Fe^{2+}$ is easily oxidized to $Fe^{3+}$, so at natural pH, iron is mostly in the form of ferric oxide and ferric hydroxide, two insoluble and very stable polymers that exist in the environment and are difficult to bioutilize. Ferriphilin, a particular iron chelating agent, meets the microbial nutrient requirements of iron by activating, absorbing, and transporting insoluble iron.

Cellulases degrade cellulose to produce glucose *via* a group of enzymes known as chitosan enzymes, which are a class of chitosan with high catalytic activity that exhibit almost no hydrolysis of chitin glycoside hydrolase; these enzymes can convert high-molecular-weight chitosan into low-molecular-weight functional chitosan oligosaccharides (*Fouda et al., 2021*). Both chitosan and cellulose are structural components of the cell walls of insects, crustaceans and fungi; thus, it can be concluded that chitinase and cellulase can breakdown the cell walls of insects and fungi (*Gürkök & Görmez, 2016*; *Maiti et al., 2017*). As mentioned in some articles, biocontrol bacteria can secrete some cell wall-degrading enzymes, which can destroy the cell wall of plant pathogens and reduce their pathogenicity. In this study, the cellulase and chitosan enzymes of TR-4 were quantitatively measured. In the Maiti study, significant similarity was detected between *B. brevis* and the M42-aminopeptidase/endoglucanase of the CelM family using high-performance liquid chromatography and mass spectrometry (*Maiti et al., 2017*).

In the inhibition of *Monilinia fructicola* by *Bacillus methylotrophicus*, *Brevibacillus* inhibited *M. fructicola*, and mycelia and spores were abnormally shaped when viewed under an SEM lens. Under TEM, the cell wall was transparent, the organelles disappeared, and the intracellular vacuoles were deformed, similar to the results of this study (*Yuan et al., 2019*). In another study, SEM revealed that *B. brevis* has antifungal, anticancer and larvicidal properties (*Fouda et al., 2022*). Like in this study, the cellulase and chitinase secreted by TR-4 decomposed the cell wall of the *C. siamense*. Therefore, the growth and pathogenicity of the *C. siamense* were inhibited.

## CONCLUSIONS

In the present study, the TR-4 strain screened from soil showed significant inhibitory effect on *C. siamense*, which was identified as *B. brevis* by 16s rRNA. Meanwhile, the experimental results showed that the inhibitory effect of 0.01 µl/ml TR-4 reached 90%, and the inhibition rate of spore germination by TR-4 on *C. siamense* reached 95%, and the relative ferritin produced 88.46 ±2.08% of ferritin, 0.2449 of glucose, and 2.15 µmol of final sugar content of chitosanase. under scanning electron microscopy and transmission electron microscopy, it was shown that TR-4 resulted in the leakage of *C. siamense* cell contents and induced cell death.

It can be concluded that TR-4 can be used as a biocontrol bacterium for more in-depth studies. This study lays the foundation for the subsequent exploration of TR-4 and provides a basis for the research and development of natural control agents.

## ACKNOWLEDGEMENTS

Thanks to Mr. Yun-fei Li for and Chi-xing Wu his guidance of the experiment, thanks to Yao Yao and Zeng-rui Zhu for their contributions to the experiment, their help in data processing and experimentation, and Sheng-feng Mao and Yan-jun Li for their support to the experiment. We thank Chi-xing Wu for his guidance of the experiment and Yan-jun Li for guidance on the experiment design and digitizing the data results.

### Funding

Funding was provided by the Natural Science Foundation of the Jiangsu Higher Education Institutions of China (22KJA220001) and the Natural Science Foundation of Jiangsu Povince (BK20231291). The funders had no role in study design, data collection and analysis, decision to publish, or preparation of the manuscript.

### Grant Disclosures

The following grant information was disclosed by the authors:
Natural Science Foundation of the Jiangsu Higher Education Institutions of China: 22KJA220001.
Natural Science Foundation of Jiangsu Povince: BK20231291.

### Competing Interests

The authors declare there are no competing interests.

### Author Contributions

- Chenxinyu Ji conceived and designed the experiments, performed the experiments, analyzed the data, prepared figures and/or tables, and approved the final draft.
- Yun-Fei Li conceived and designed the experiments, authored or reviewed drafts of the article, and approved the final draft.
- Yao Yao performed the experiments, analyzed the data, prepared figures and/or tables, and approved the final draft.
- Zengrui Zhu performed the experiments, analyzed the data, prepared figures and/or tables, and approved the final draft.
- Shengfeng Mao conceived and designed the experiments, authored or reviewed drafts of the article, and approved the final draft.

### DNA Deposition

The following information was supplied regarding the deposition of DNA sequences:
The 16S rRNA sequences of TR-4 are available at GenBank: OP658963.

## Data Availability

The raw data is available in the Supplemental Files.

## Supplemental Information

Supplemental information for this article can be found online at http://dx.doi.org/10.7717/peerj.17568#supplemental-information.

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
