# Peer review of "Mechanism of Brevibacillus brevis strain TR-4 against leaf disease of Photinia×fraseri Dress"

_PeerJ, doi:10.7717/peerj.17568_

## Round 0.1 · original submission · Major Revisions

Dear Author

Serious criticisms were made by reviewers regarding the manuscript's language and structure. Request you to revise your manuscript in light of the reviewer's suggestions.

with regards

**Language Note:** The review process has identified that the English language must be improved. PeerJ can provide language editing services - please contact us at [email protected] for pricing (be sure to provide your manuscript number and title). Alternatively, you should make your own arrangements to improve the language quality and provide details in your response letter. – PeerJ Staff

Reviewer 1 ·

Basic reporting

The authors identified Brevibacillus brevis and its inhibitory effect against leaf disease caused by
Colletotrichum, meanwhile, the authors also investigated the possible mechanism and interaction between B. brevis and the pathogen. The topic of the manuscript is of importance as biocontrol is one of the most favourable way to control disease in practice. However, the manuscript needs major improvement before it can be published. Main concerns include: 1. English languge does not match professional criteria for scientific paper. Many unfinished sentences, too many grammar mistakes....2. the manuscript structure was wrong. Result and Discussion section without any references! Conclusion section is far too long! 3. The entire manuscript was lacking of logic. Explain the purpose of each section. 4. the table and figure legends were not correct! supplementary files without any legends!

Experimental design

The experimental design was not described clear, especially lack of logic.

Validity of the findings

Conclusion section is far more too long!

Reviewer 2 ·

Basic reporting

I believe the authors have a valid approach to conduct their study and identify potential biocontrol agents for Colletotrichum. Unfortunately, I couldn't thoroughly review the manuscript because of some lack of clarity. The English language should be improved to ensure that an international audience can clearly understand the text. Some examples where the language could be improved include lines 47-51, 125-126 – the current phrasing makes comprehension difficult. I suggest you have a colleague who is proficient in English and familiar with the subject matter review your manuscript, or contact a professional
editing service.

I would suggest to improve the structure of the introduction as well, as currently the last two paragraphs go back and forth between what is already known and what the study will do, and the line between the two is unclear.

There are some misused terms such as "soil endophytic bacteria" line 58 (there are some plant endophytic bacteria, or soil bacteria but there is no such thing as soil endophytic bacteria), or "TR-4 endocrine enzyme" line 137 (I have never heard of the term endocrine used to refer to bacteria). Some virus culture are mentioned line 191 but I do not think there are any virus in this study.

Experimental design

For what I could guess, I think the experimental design and methods used are appropriate and robust, but the text really lack some clarity as well as key details of the experimental procedure to allow me to properly review the work. For instance, line 104, what are the appropriate controls? Line 109, what does EC stands for? I don't understand at all the experiment here. Nothing is said on bacterial concentration in the in planta tests.

Validity of the findings

Again, I cannot assess the validity of the findings because of the lack of clarity of the text

Reviewer 3 ·

Basic reporting

Review for Chenxinyu Ji et al

The article focuses on screening biocontrol bacteria from the rhizosphere soil of Photinia x fraseri Dress to combat a leaf disease caused by Colletotrichum. The study identifies a strain named Brevibacillus brevis TR-4, which shows strong antagonistic effects against Colletotrichum, achieving a 98% inhibition rate. As confirmed by electron microscopy, TR-4's mechanism involves degrading the pathogen's cell wall. The research underscores the potential of TR-4 as a biocontrol agent in agriculture, highlighting its significance in managing plant diseases and reducing economic losses.

The articles lack an engaging writing style. The articles need improvement in English writing at serval places.

Line 28 Rephrase the sentence.

Photinia x fraseri Dress is a small evergreen tree or shrub that grows well in warm, moist
environments(Toscano, Farieri, Ferrante, & Romano, 2016). But under direct light, its color is more vivid, mainly distributed in southeast Asia and eastern Asia and North America subtropical and temperate areas, in many provinces of China has also been widely cultivated, as a mass planting of garden greening tree, its disease will bring a large number of economic losses (Li et al., 2023).

To

"Photinia x fraseri Dress, a small evergreen tree or shrub, prefers warm, moist environments and exhibits more vibrant colors in direct light (Toscano, Farieri, Ferrante, & Romano, 2016.) Predominantly found in Southeast Asia, Eastern Asia, and North America, it is widely cultivated across various Chinese provinces for garden greening. The plant's disease susceptibility has significant economic implications (Li et al., 2023)."


Line 30 Remove extra space after more vivid, mainly…...


Line 37 disease was identified as Colletotrichum in a genus……….
This sentence does not make any sense to me. You can rephrase it to

the disease affecting the plants was identified as being caused by the Colletotrichum genus of fungi. This genus is known for attacking the roots, stems, leaves, flowers, and fruits of various plants globally, leading to decreased agricultural product quality and significant economic losses. Cao et al., 2021."

Line 45 what are bio based drugs?

Line 183 italicize in vitro.

Line 202 italicize in vivo.

Line 239 What is chitanase is it a new enzyme? or is it chitinase?

Figure 1 Add a high-resolution image. The current image is very blurry.

Figure 1 and Figure 2 add scale bars to both figures.

Figure 3 legends. Italicize Colletotrichum siamense.

Figure 6 I do not see the length of the scale bar in A and B.

Experimental design

no comment

Validity of the findings

no comment

Additional comments

The article focuses on screening biocontrol bacteria from the rhizosphere soil of Photinia x fraseri Dress to combat a leaf disease caused by Colletotrichum. The study identifies a strain named Brevibacillus brevis TR-4, which shows strong antagonistic effects against Colletotrichum, achieving a 98% inhibition rate. As confirmed by electron microscopy, TR-4's mechanism involves degrading the pathogen's cell wall. The research underscores the potential of TR-4 as a biocontrol agent in agriculture, highlighting its significance in managing plant diseases and reducing economic losses.

---

## Round 0.2 · Major Revisions

Reviewers have suggested major revisions. Kindly follow their given comments and revise the manuscript accordingly.

**Language Note:** The review process has identified that the English language must be improved. PeerJ can provide language editing services - please contact us at [email protected] for pricing (be sure to provide your manuscript number and title). Alternatively, you should make your own arrangements to improve the language quality and provide details in your response letter. – PeerJ Staff

Reviewer 1 ·

Basic reporting

1. The revised manuscript did not use clear and professional English. There are too many language mistake all over the manuscript, including basic grammar, wrong expression....e.g., line 20: 'the isolates were ' should be 'the isolate was'; line 23: 'effects of TR-4 on leaves', what does this mean? Line 26: 'pathogenic bacteria' ? Line 56:"PGPR' ? Line 64: 'a series of'? Line66: 'the TR-4 PAIR'? Line 67: 'later period'? Line 72: 'strain strain'?
2. As pointed out in the first version, the structure of the manuscript is not well-organised and need to be improved. The material and method section has to be combined and make more logic and concise. The result and discussion section needs to be re-organized!
The figure legend and table caption are not correct.

Experimental design

The experiment design is basically ok. However, the methods used were described badly. E.g.,line 136-149. More importantly, the authors seem to have wrong understanding on the rhizosphere microbe: the rhizosphere bacteria was mentioned all over the manuscript, however, the line 80-83 indicated that the bacteria were isolated from bulk soil rather than rhizosphere soil !!!!

Validity of the findings

The results section were organised in a very bad way. Some contents were not results at all, e.g., line 178-184; line 236-240!! The results and discussion were mixed together, which makes this section difficult to follow. The discussion content should be related to the main results obtained and should not be general knowledge, e.g., line 187-192, 274-285.......

Reviewer 2 ·

Basic reporting

The authors used a professionnal service to improve the language of the manuscript and I did note a great improvement. However, there are still too many information missing, and words innappopriately used. This is something which cannot be corrected by a third party and the authors or colleagues understanding the meaning of the text need to carefully review and edit the manuscript. Below are some exemples of misused words and missing information. This is not an exhaustive list as this was too much work for me to go through all the mistakes in the writing but hopefully this will provide some guidance.

Misused terms:

- L89 'plate labeling'. I don't think plate labelling is a way of isolating bacteria
- L93 'bacteriostatic activity' you are measuring the inhibition of a fungi. So this is not a bacteriostatic activity
- L115 'Inhibitory effect of TR-4 on pathogenic bacteria' as far as I understood you are working on a pathogenic fungi, not a bacteria. Same goes for L157
L198-L199 'to determine the effect of pesticide' the study is not testing any pesticide!
- 208-209 'fermentation filtrate' did you really have some fermentation happening there?

Missing informations:
- L80 Where did the plants come from?
- L98 How was the growth inhibition calculated?
- L101 What is the 'strain solution'? Is it a sterilised culture supernatant?
- L105 For how long did the experiment run?
- L109 how much cell per mililiter does an OD600 of 5 represents?
- L112 this is the first mention of anthrax! Which species are we talking about? Is is C. siamense?
- L118 How were the spores inoculated?
- L222 What is the sample size here? How many plants? How many leaves?
- L128 How was the DNA extracted? Which primer concentration?
- L137-149 You describe how the method works here. You should explain how you did the experiment instead.

Lack of clarity:
- L68-70
- L84 from 1 to 10 what? I don't understand this
- L90 after counting what? No counting was reported in the resutls
- L102-103 What is the median effective concentration? This whole part still does not make sense to me. How was EC50 determined?

Undefined abbreviations:
- L140 CAS
- L160 CK

Experimental design

In this version of the manuscript could understand a little bit better the experimental design and I think the science conducted here is valid. However there are still a number of missing in formation (see "Basic reporting" section).

Validity of the findings

I think the findings are valids.

Reviewer 3 ·

Basic reporting

Upon review of your revised manuscript, it has become evident that the revisions submitted fall significantly short of the required standards and fail to address the critical feedback provided. The absence of a detailed, point-by-point response to the reviewers' comments is unacceptable and disregards the standard academic protocol for manuscript revisions.
Furthermore, the document's heavily edited state, without a clear indication of changes made, severely impedes any meaningful review process. Such a submission not only shows a lack of respect for the review process but also suggests a disregard for the value of constructive feedback.
Additionally, the omission of figures and essential visual content further complicates the review process, making it impossible to fully assess the merits of your research. This oversight must be rectified immediately.
Given these significant shortcomings, I must insist on the following actions:
1. Comprehensive Revision: Submit a thoroughly revised manuscript that clearly addresses each of the reviewers' comments in a detailed, point-by-point manner. This revision must leave no comment unaddressed.
2. Document Clarity: Provide a clean version of the manuscript with all changes highlighted or tracked.

3. Inclusion of Visuals: Ensure that all figures and visual content are included in the revised submission.

I believe these steps are crucial for a thorough and fair assessment of your work. Please let me know if you have any questions or require further clarification on any of the points mentioned.
Thank you for your understanding and cooperation. I look forward to your response and the revised manuscript.

Experimental design

'no comment'

Validity of the findings

'no comment'

---

## Round 0.3 · Minor Revisions

The reviewer's report was received, and one of the reviewers suggested a minor revision. Kindly read the reviewer's comments and improve the manuscript accordingly.

Reviewer 1 ·

Basic reporting

The authors claimed that the manuscript has been edited by professional english editing. However, there are still many mistakes in the text in terms of not clear english, wrong use of words and sentences.

Secondly, it seems the authors did not understood the comments from the reviewer. Many comments have not been revised correctly.
For example, the article structure in material and method section: several section should be combined and make this section more concise and logical. It is not just number each section in Material and Method, which is actually wrong in the format of the manuscript. Additionally, the conclusion section should be summarise the main results, rather than repeating general knowledge and results. The current conclusion section is far more too long and is not conclusion!

The pervious comments: line 23: 'effects of TR-4 on leaves', what does this mean? The sentence was wrong meaning!. TR-4 is targeting the pathogen, not the plant leaves.....

What does the pictures in the figure 1? what the authors try to show?...
The titles of figures should be concise and clear, which has not been revised...

Experimental design

no materials and methods section needs to be re-arranged for concise and logic.

Validity of the findings

conclusion are not stated!

Additional comments

no comments

Reviewer 3 ·

Basic reporting

I appreciate the authors for their prompt and thorough revisions addressing my earlier concerns. The modifications have significantly strengthened the manuscript, and I am now satisfied with the completeness and clarity of the content.

Experimental design

no comment

Validity of the findings

no comment

Additional comments

no comment

---

## Round 0.4 · Minor Revisions

One of the reviewers has suggested final very minor revisions. Please have a look at the comments given below and revise the manuscript.

Reviewer 1 ·

Basic reporting

Dear Editor,

The manuscript has been revised by the authors following the comments suggested. The manuscript has been improved significantly compared to the previous version. The manuscript can be accepted after minor revision on the conclusion section. In my opinion, the conclusion should be more concise and clear summary of the main result, rather than repeating the main result. Secondly, the conclusion is far more too long.

Best regards,
NA
Hui

Experimental design

NA

Validity of the findings

NA

---

## Round 0.5 · accepted · Accept

Thank you for submitting your revised manuscript to PeerJ. I am pleased to accept your manuscript for publication in its current form. This decision is based on the advisor's report and their consent, indicating that the authors have addressed all comments raised during the review process.

Reviewer 1 ·

Basic reporting

The author has made all necessary revision according to the comments and suggestions, and the manuscript has been improved significantly. Therefore, it can be accepted.

Experimental design

NA

Validity of the findings

N/A